# SMART (artificial intelligence enabled) DROP (diabetic retinopathy outcomes and pathways): Study protocol for diabetic retinopathy management

Padmaja Kumari Rani[1,2]*, DurgaBhavani Kalavalapalli[3], Raja Narayanan[2], Shyam Kalavalapalli[3], Ritesh Narula[2,4], Rakesh K. Sahay[3,5], Sarang Deo[6]

**1** Department of Teleophthalmology, L V Prasad Eye Institute, Hyderabad, Telangana, India, **2** Anant Bajaj Retina Institute, L V Prasad Eye Institute, Hyderabad, Telangana, India, **3** IDEA (Institute of Diabetes, Endocrinology, and Adiposity) Clinics, Hyderabad, Telangana, India, **4** LILAC (L V Prasad eye institute Image Laboratory and Analysis Centre), Hyderabad, Telangana, India, **5** Osmania General Hospital, Hyderabad, Telangana, India, **6** Max Institute of Health Care Management, Indian School of Business, Hyderabad, Telangana, India

* rpk@lvpei.org

## Abstract

### Introduction

Delayed diagnosis of diabetic retinopathy (DR) remains a significant challenge, often leading to preventable blindness and visual impairment. Given that physicians are frequently the first point of contact for people with diabetes, there is a critical need for integrated screening programs within diabetes clinics to enhance DR management and reduce the risk of severe vision loss.

### Methods and analysis

We will conduct a prospective cohort study comparing (i) the intervention cohort, screened at diabetes clinics and referred to eye clinics per the proposed pathway, and (ii) the standard-of-care (SOC) eye clinic cohort. The study will be conducted in Hyderabad, India, at LV Prasad Eye Institute and four IDEA (Institute of Diabetes, Endocrinology, and Adiposity) Clinics. The primary objective is to evaluate the effectiveness of a systematic diabetic retinopathy screening program in achieving earlier detection and reducing visual impairment among People With Diabetes (PWD) attending IDEA clinics compared to routine care at eye care settings. The screening program will be operationalized using AI-enabled tools and supported by trained non-medical technicians. We will perform visual acuity tests and non-mydriatic fundus photography using AI-assisted cameras. DR-positive patients will be referred for treatment and follow-up. We aim to achieve high accuracy (>90%) in appropriate referral of DR and high screening coverage (>80%) of eligible PWD. Success metrics

**Data availability statement:** No datasets were generated or analysed during the current study. All relevant data from this study will be made available upon study completion.

**Funding:** RPK received the funding . Grant number: IIRP-2023-4272/F1 Name of the Funder:ICMR—Indian Council of Medical Research) Website: https://epms.icmr.org.in/ Funders did not play any role in the study design, data collection and analysis, decision to publish, or preparation of the manuscript

**Competing interests:** The authors have declared that no competing interests exist.

include screening uptake, AI diagnostic accuracy, referral rates, cost-effectiveness, patient satisfaction, follow-up adherence, and long-term outcomes.

## Conclusion

This study aims to enhance diabetic retinopathy screening and management through an AI-enabled approach at diabetes clinics, improving early detection and care pathways. The findings will contribute to evidence-based strategies for optimizing DR screening and management, with results disseminated through peer-reviewed publications to inform policy and practice.

## Trial registration

**Trial registration number:** CTRI/2024/03/064518 [Registered on: 20/03/2024] (https://ctri.nic.in/).

## Introduction

Diabetes Mellitus (DM) is a major public health problem worldwide, affecting over half a billion people (537 million in 2021), with projections indicating an increase to 783 million by 2045. Half of these individuals reside in the USA, China, and India [1]. In 2023, it is estimated that India is home to 101.3 million people with diabetes mellitus (DM) and 136.0 million people with pre-diabetes [2]. Diabetic retinopathy (DR) is a significant microvascular complication of DM, caused by damage to the blood vessels in the retina, leading to vision loss or blindness. The prevalence of DR is increasing due to the rising incidence of diabetes, making it a leading cause of blindness globally and in India [3].

A nationwide multicentric study in India found the prevalence of DM in adults aged 40 years and older to be 18%, DR at 12.3%, and vision-threatening diabetic retinopathy (VTDR) at 4%, estimating 3 million people with VTDR in India [4]. The estimated national prevalence of vision impairment among people with known or undiagnosed diabetes was 21.1% (95% CI (confidence intervals) 15.7–27.7) and blindness was 2.4% (1.7–3.4). A higher prevalence of any vision impairment (30%) and blindness (6.7%) was observed in those with ungradable images (indicating cataract burden). Among known diabetics, vision-threatening diabetic retinopathy and diabetic macular oedema were associated with blindness. Based on the estimated 101 million people with diabetes in 2021, approximately 21 million people in India have vision impairment due to diabetes, of whom 2.4 million are blind, with a higher prevalence observed in those from lower socioeconomic strata [5]. Landmark trials over decades have shown that early detection and treatment of DR can prevent or delay its progression and reduce the risk of vision loss [6–8]. For people with diabetes aged 40 years or above, annual screening followed by eye examinations would cost around 42.3 billion Indian rupees (INR) per year; treating sight problems around 2.87 billion INR per year if 20% of those needing treatment receive it; and lost economic activity around 472 billion INR. Moreover, 2.86 million quality-adjusted life years (QALYs) are lost annually due to blindness and moderate-severe visual impairment (MSVI) [9].

The effectiveness of systematic DR screening programs in reducing blindness due to DR has been established in high-income countries [10]. Targeted telemedicine-based screening and referral of people with diabetes have been shown to be more effective than universal referral for DR screening in India [11,12]. However, there is a lack of large-scale systematic screening initiatives in low- and middle-income countries like India [13]. India's complex and fragmented healthcare delivery system presents a spectrum of contrasting landscapes, including socioeconomic factors, issues with medical infrastructure, insufficiencies in the supply of medical requisites, and economic disparities in diabetes care delivery between government and private sectors [14,15]. The absence of systematic screening programs in India results in many undiagnosed cases of DM and DR, leading to delays in treatment. The lack of access to timely treatment can cause irreversible vision loss and increase the burden on the healthcare system. The situation is compounded by the lack of awareness of DR among people with diabetes and the limited availability of ophthalmologists in many parts of the country [16,17]. With a ratio of 18 ophthalmologists per million population, each ophthalmologist in India is required to examine 3500 people with diabetes annually [18,19]. The actual number would be higher due to several nonpracticing ophthalmologists and people at risk of developing diabetes. Additionally, there are only 1100 registered retinal specialists [20].

To address this problem, effective screening programs are needed to detect DR early and provide timely treatment. Implementing such programs requires a collaborative effort from healthcare providers, policymakers, and the community. Innovative technologies, such as telemedicine and artificial intelligence, could significantly improve the detection and management of DR in low-resource settings [21,22]. Providing one-stop care at a diabetes clinic for complications of diabetes, including DR, can also help address the problem by offering comprehensive care that includes screening, diagnosis, and treatment for DR and other complications of diabetes, ensuring timely and coordinated care that can improve outcomes and reduce the burden on the healthcare system.

Physicians play a pivotal role in screening for diabetic retinopathy (DR), as they are often the initial point of contact for individuals with diabetes. Integrating DR screening within a physician's office has proven to be more effective than traditional self-reported screening models [11,23]. To effectively manage Diabetic complications like DR, implementing a systematic DR screening program at the physician clinic level is an optimal strategy. Fundus photography-based models are particularly effective for DR screening, as they allow for objective assessment [12,24]. Utilizing non-mydriatic fundus cameras with built-in artificial intelligence algorithms in physician offices enables trained technicians to capture images efficiently.

In India, established guidelines for DR screening are in place, and a recent positive development is the inclusion of DR screening in the Ayushman Bharat Pradhan Mantri Jan Arogya Yojana (AB PMJAY) [25,26]. Under this program, both ophthalmologists and general physicians are eligible for reimbursement for DR screening services. The literature offers insights into DR screening guidelines [27,28], the feasibility of telemedicine referral pathways between diabetes and ophthalmology clinics [12], and the inclusion of DR screening reimbursement under AB PMJAY [29]. However, there remain gaps in evidence, particularly regarding disease staging, operational and clinical outcomes of subsequent treatment and follow-up, and the cost-effectiveness of the referral pathway for both healthcare providers. The objective of the present study is to address these gaps and optimize strategies for DR screening and management.

## Study hypothesis

This study aims to investigate whether implementing a systematic DR screening program at the physician clinic can lead to earlier detection of blindness and visual impairment caused by DR compared to routine DR care pathways at a tertiary eye care setting. As the physician is typically the first point of contact for people with diabetes, implementing such a program may improve the overall management of DR and reduce the risk of severe vision loss. This study could help in the development of more effective DR screening and management strategies, improving outcomes for people with diabetes.

 

## Objectives

The study aims to show that systematic screening leads to earlier detection of diabetic retinopathy (DR) and reduces visual impairment. It will compare the effectiveness of screening at physician clinics versus routine DR care in eye care settings. The screening program, supported by trained non-medical technicians, is AI-enabled and targets over 90% accuracy in appropriate referral of DR and 80% coverage of eligible people with diabetes (PWD).

The secondary objectives of the study are to: 1. Assess the cost-effectiveness of the AI-enabled systematic screening program and referral pathway compared to the current pathway at an eye care clinic. 2. Evaluate patient satisfaction and adherence to follow-up recommendations in the AI-enabled screening program versus the routine care pathway through an effective digital tracking system.

## Methods and analysis

### Study design

We will conduct a prospective cohort study involving a comparison between two patient groups: (i) intervention cohort: patients screened at the diabetes clinics using AI and trained nonmedical technicians and referred to the eye clinic as per the proposed pathway, and (ii) standard-of-care (SOC) cohort: patients who arrive at the ophthalmology clinic as per current practice - either self-walk-ins or referrals from other healthcare providers. Fig 1 shows SPIRIT template for clinical trial.

### Study setting

All the study sites will be in the city of Hyderabad in Telangana, India (study area map: Fig 2)

•LV Prasad Eye Institute (Kallam Anji Reddy Campus at Banjara Hills) (Tertiary eye clinics)

•IDEA Clinics (Madhapur, Kondapur, Ameerpet, Kukatpally Housing board (KPHB) (Diabetes clinics)

The LV Prasad Eye Institute is a leading tertiary eye care center with advanced infrastructure for diabetic retinopathy diagnosis and treatment, making it an ideal reference for validating the intervention. The IDEA Clinics are strategically located in urban areas with a high burden of diabetes and diabetic retinopathy, ensuring the recruitment of a relevant and representative population. These clinics provide comprehensive diabetes care and offer an optimal setting to test the integration of AI-enabled diabetic retinopathy screening within existing care pathways. The combination of these sites allows for the evaluation of the intervention's feasibility and scalability in diverse healthcare settings, supporting its potential for broader implementation.

The Indian School of Business team will oversee project management, research methodology, data analysis, and cost-effectiveness analysis, leveraging their management and research expertise to enhance the collaboration.

### Selection of participants

The inclusion criteria for this study in the intervention arm are People with Diabetes (PWD) who visit the diabetes clinics who have never had a fundus examination or fundus photo screening IDEA clinics (Diabetes clinics)/LVPEI(Eye clinics). The referral source will be captured as self-reported, physician, LVPEI network, or ophthalmologist in the standard of care arm -LVPEI eye clinic pathway.

The exclusion criteria for the study are designed to omit individuals with previously diagnosed Diabetic Retinopathy (DR) at the time of presentation to diabetes clinics during the study period. Although we will not include the images of these patients in the study analysis, we will offer appropriate care for their DR at the eye clinic. Additionally, we will seek consent from all eligible participants before they are enrolled in the study and subjected to study protocol. Participants enrolled in the project will be between 18 and 90 years of age.

# SPIRIT Template for Clinical Trial

This document maps the clinical trial details to the SPIRIT schedule of enrollment, interventions, and assessments.

## Study Schedule of Enrollment, Interventions, and Assessments

| Study Period | Enrolment | Allocation | Post-allocation | Close-out |
|---|---|---|---|---|
| Timepoint | March 29, 2024 (Enrollment start) | Individual-specific enrollment date (Day 0) | +3 months, +6 months, +12 months | End of study (April 2026) |
| Eligibility screening | Screening participants aged 18–90 years for diabetes diagnosis and absence of prior DR treatment | x | x | x |
| Informed consent | Written informed consent obtained after participants are briefed about the study goals, AI screening methods, and follow-up protocols. | x | x | x |
| [Other procedures] | x | x | x | x |
| Allocation | x | Allocated to an AI-enabled DR screening group at diabetes clinic/routine care pathway at eye clinic based on diabetes diagnosis and absence of prior DR treatment. | x | x |
| Intervention A | Diabetes diagnosis and | Visual acuity and | Regular follow-ups at 3, 6, and 12 | Final data collection for |

**Fig 1. SPIRIT template for clinical trial.**

| | | | | |
|---|---|---|---|---|
| | absence of prior treatment or diagnosis of DR | nonmydriatic fundus photography with AI-enabled DR screening and followup | months based on DR diagnosis | primary and secondary outcomes. |
| Comparator (Standard care) | Diabetes diagnosis and absence of prior treatment or diagnosis of DR | Patients presenting via self-referral or non-systematic pathways undergo retinal exams and fundus photography. | Regular follow-ups at 3, 6, and 12 months | Final data collection for primary and secondary outcomes. |
| Baseline variables | Demographics, medical history, | x | x | Final Data analysis |
| Outcome variables | The primary objective is to evaluate the effectiveness of a systematic diabetic retinopathy screening program in achieving earlier detection and reducing visual impairment among People With Diabetes (PWD) attending IDEA clinics compared to routine care at eye care settings. | 1.DR detection rate and STDR detection rate between the arms 2.Screening uptake rate 3.Sensitiviy and specificity of AI for diagnosis and referral rate Timely referral rates | | Final data collection for primary and secondary outcomes. |
| Other data variables | 1. Assess the cost-effectiveness of the AI-enabled systematic screening program and referral pathway compared to | x | Cost effective analysis (Direct and Indirect expenses) Patient satisfaction levels, Adherence to followup recommendations | x |
| | the current pathway at an eye care clinic. 2. Evaluate patient satisfaction and adherence to follow-ups via digital tracking. | | | |

**Fig 1.** Continued.

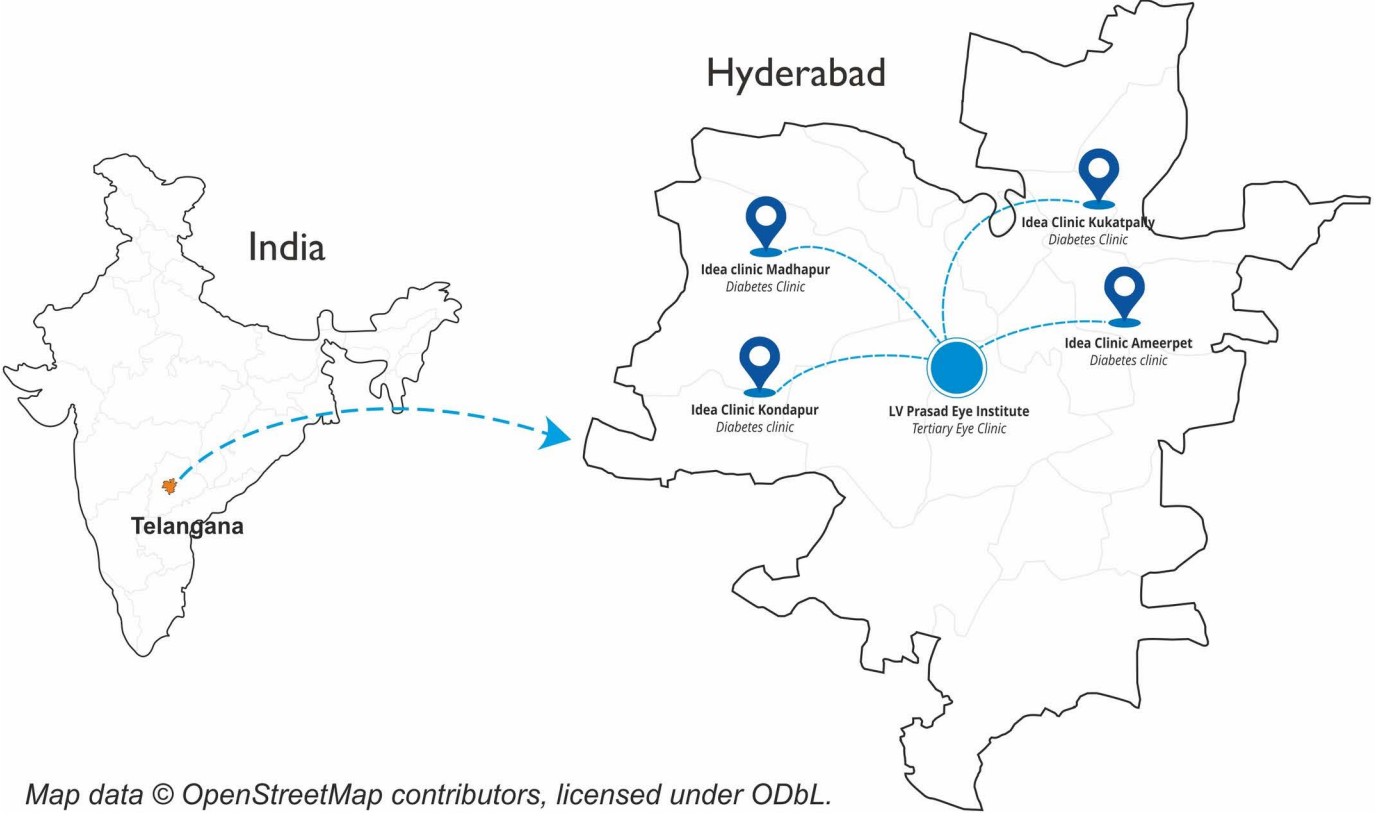

Hyderabad

India

Telangana

Idea clinic Madhapur
*Diabetes Clinic*

Idea Clinic Kukatpally
*Diabetes Clinic*

Idea Clinic Ameerpet
*Diabetes clinic*

Idea Clinic Kondapur
*Diabetes clinic*

LV Prasad Eye Institute
*Tertiary Eye Clinic*

*Map data © OpenStreetMap contributors, licensed under ODbL.*

**Fig 2. Study area map.**

## Study procedures

The study procedures are detailed in the study flow chart (refer to Fig 3).

We will invite eligible patients for screening through a call and recall system. Visual acuity in both eyes will be recorded using a tablet with a web-based vision checking application (Peek Vision). Non-mydriatic fundus photography of both eyes will be performed using AI-integrated retinal cameras (Forus, 3 Nethra Classic HD with AI Integration in two facilities, and Remidio, Fundus on Phone with AI integration, in two facilities). These AI tools have undergone internal validation and published by the company prior to its integration into the cameras [30,31]. Our project, we will be conducting a real-world validation to assess the performance and effectiveness of this AI tool in practical clinical settings.

Diabetic retinopathy grading will be conducted as per International Clinical Diabetic Retinopathy severity scale [32]. Sight-threatening diabetic retinopathy (DR) is defined as severe non-proliferative diabetic retinopathy (NPDR), proliferative diabetic retinopathy (PDR), and moderate to severe diabetic macular edema (DME). The referral criteria include any patient with DR, visual acuity less than 20/60, any other retinal pathology requiring referral, or ungradable retinal images, or any combination of these factors.

In the AI-enabled screening process for Diabetic Retinopathy (DR) using fundus images, if no signs of DR are detected by the AI, the patient will receive a label of "no DR diagnosis" at the initial stage. However, if DR is indicated, a trained technician at the Idea Clinic will manually review the images to confirm or rule out DR, and the result will be communicated to the patient by the physician. Simultaneously, all images will undergo further manual grading at the Central Image

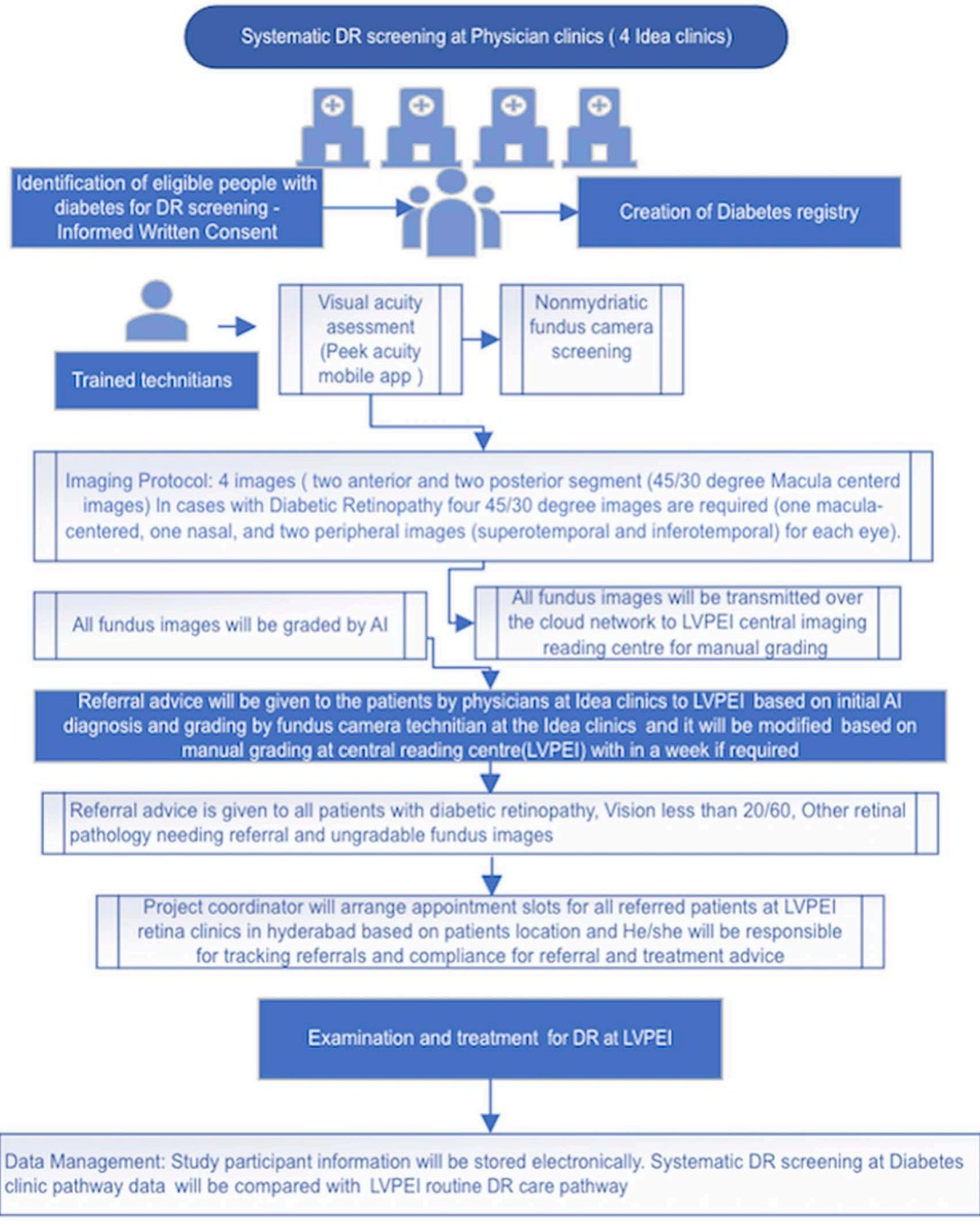

**Fig 3. Study flow chart.**

Reading Center at LVPEI for quality assurance. If this manual grading results in a change in diagnosis, the updated result will be shared with the patient within a week since their initial screening.

The images captured at each physician clinic facility will be transferred through a cloud-based teleophthalmology system to the LVPEI central image reading centre (LILAC (L V Prasad eye institute Image Laboratory and Analysis Centre),

where grading will be done by two trained graders. Final grading of all referable cases of DR, ungradable images, other pathology needing referral, and an audit of 5% of normal fundus images will be conducted by an experienced ophthalmologist. Patients diagnosed with DR will be referred for appropriate treatment and provided with follow-up advice. Follow-up appointments will be arranged at LVPEI.The follow-up periods for individual participants will be based on the diabetic retinopathy diagnosis and scheduled at 3, 6, and 12 months intervals.

## Study timeline

Following the approval of our clinical trial registration by the Clinical Trials Registry of India (CTRI) (https://ctri.nic.in) on March 20, 2024, we commenced patient recruitment on March 29, 2024. Currently, recruitment and data collection are ongoing at both the intervention arm (Idea Clinics) and the standard-of-care arm (LV Prasad Eye Institute). An interim data analysis is scheduled for one year from the recruitment start date. The total duration of the project is three years.

## Quality assurance, quality control measures

To ensure rigorous quality assurance and control, the study will include comprehensive technician training with quarterly refreshers and bi-monthly calibration of AI screening equipment. Comprehensive training programs will be provided to non-medical technicians, focusing on the correct use of AI-assisted screening equipment, standardized procedures for visual acuity tests, and accurate capture of fundus photographs. Technicians at diabetes clinics will be trained in visual acuity estimation and validation tests (n = 25) to ensure reliability and accuracy. The visual acuity tool used in the study (Peek Vision) has been globally validated in multiple studies and effectively utilized by technicians and field workers in diverse settings [33,34]. The sample size of **N = 25 patients (50 eyes)** is intended for pilot testing following standardized training. This pilot will serve as an initial assessment of the reliability and accuracy of visual acuity estimation using the mobile VA application by trained technicians. The pilot testing results will be used to validate the training protocol and identify any gaps requiring refinement.

To ensure ongoing reliability, periodic audits will be conducted during training refresher programs, where technicians' performance will be re-evaluated. Additionally, comparisons of visual acuity estimates from referred patients will be performed to assess consistency over time. This iterative approach ensures that the quality of visual acuity measurements is maintained throughout the study. They will also receive training in fundus image capture, study image protocol, DR grading and referral criteria.

Quality control measures will be implemented to monitor and verify the accuracy of DR and STDR detection rates, timely referral rates, and other key metrics. A quality assurance process will compare AI grading with manual grading by the technician and the Central Reading Center to ensure screening accuracy and reliability.

Weekly audits will validate 5% of normal fundus images, review all DR and retinal pathology, ungradable cases, and ensure adherence to screening protocols. Monthly physical visits to diabetes clinics will monitor protocol compliance, and both weekly team reviews and monthly investigator meetings will assess project performance and outcomes. Patient outcomes will be cross-referenced with established benchmarks (e.g., National DR screening guidelines) to verify the accuracy of DR detection.

## Data management

We will centrally store all study data on a cloud-based data management portal to improve data accuracy and integrity. All data fields will be entered using online spreadsheets to enable seamless entry from both physician and eye clinic sites. A robust data management system will be employed to handle the large volume of data generated, ensuring patient confidentiality and compliance with data protection regulations. The system will facilitate seamless data entry, real-time monitoring, and efficient retrieval of information and tracking of all patients with DR for analysis.

To avoid manual data entry and the need to manage multiple storage platforms and physical consent forms, we will roll out the portal in phases. The first phase will focus on creating a diabetes registry that automates 80% of fields from the IDEA Clinics Electronic Health record (HER) and enables physicians to nominate patients for the Smart DROP Project with a single click from the EHR prescription module. The second phase will introduce digital consent forms and integrate fundus image handling for easier storage and access. The third phase will streamline the Case Proforma, and the portal will also include data visualization through reporting and analytics, as well as communication tools like visit reminders for enrolled patients. The portal will be accessed by designated project users with secure usernames and passwords to ensure confidentiality and data security.

## Statistical analysis

Accuracy of AI-enabled cameras: Taking the grading of images done by the experts as the ground truth, we will estimate the accuracy of the AI- enabled cameras (operated by a non- ophthalmic technician) using standard measures such as sensitivity, specificity, and area under the ROC (Receiver Operating Characteristic). Multivariable regression models will be employed to adjust for potential confounders, including age, diabetes duration, baseline visual acuity, and other relevant factors. These models will allow for a more accurate assessment of the relationships between the intervention and key outcomes, such as disease staging at diagnosis, referral rates, and long-term visual outcomes, by accounting for the influence of these confounders. Both linear and logistic regression models will be used, depending on the nature of the outcome variable, to ensure robust and reliable results. Additional statistical tests, such as chi-square tests for categorical outcomes and t-tests or ANOVA for continuous outcomes, will be employed as appropriate.

## Sampling

The study will employ a **convenience sampling method**, recruiting all consecutive eligible individuals with diabetes mellitus (DM) visiting physician clinics and eye care settings who meet the standard inclusion and exclusion criteria and provide informed consent. While convenience sampling is utilized, potential sources of bias include **selection bias, site-specific variation, differential access to care and Observer Bias**. To mitigate these, systematic recruitment will be ensured by enrolling all eligible patients consecutively at each site, reducing the likelihood of preferential selection. Recruitment will take place across multiple physician clinics and eye care settings to improve representativeness and minimize site-specific variation. Differential access to care will be monitored by assessing demographic and clinical characteristics of enrolled participants, allowing adjustments to recruitment strategies if needed. To prevent observer bias, standardized screening protocols and training for healthcare personnel will be implemented, ensuring consistency in recruitment and data collection. These measures aim to enhance the validity and generalizability of the findings while acknowledging the inherent limitations of convenience sampling.

## Sample size

We calculate the sample size to detect meaningful differences in the disease staging of patients between two arms at their first eye consultation. Based on the historic data from LVPEI, we assume a 50% prevalence of STDR among DR patients in the SOC arm. Also, we assume that the prevalence of STDR in the intervention arm at the time of screening in the diabetes clinic would be much lower (approximately 10%). However, patients in early stages of DR may not comply with the referral and hence the prevalence of STDR among patients who complete their first ophthalmology visit may be higher (approximately 25%). Based on these estimates, to achieve a power of 80% with a Type-I error rate of 5%, we estimate a sample size of 60 patients in each arm. Since the additional intervention required for the study is minimal (a short check-up using a camera), we expect that about 80% of the eligible patients will consent to participate. Thus, we will need 75 patients with DR to be identified in the intervention arm. Further, based on the historic data at the IDEA clinics,

we estimate that 20% of all new registering diabetic patients will have DR. Thus, we will plan to register and screen 375 new diabetic patients each at IDEA clinics (intervention) and LVPEI clinics (SOC arm).

To evaluate the AI tool's diagnostic accuracy with sensitivity and specificity set at 80%, a sample size of 246 DR-positive cases is required for sensitivity analysis, and 246 DR-negative cases for specificity analysis, assuming a 95% confidence level and a margin of error of ±5%.Given the estimated 20% prevalence of diabetic retinopathy (DR) in the screened population, a total of 1230 patients will be screened to ensure adequate representation of both DR-positive and DR-negative cases.

To assess patient satisfaction, a sample size of **316 patients** will be selected from both the AI-enabled and routine eye care pathways, calculated to detect a difference in satisfaction rates (80% vs. 65%) with a **95% confidence level** and a **margin of error of ±5%**. Random sampling will be employed to ensure unbiased selection of participants from each pathway. This method will involve assigning unique identifiers to eligible patients and using a randomization tool to select participants, ensuring the sample is representative of the study population.

Patient satisfaction will be assessed using a structured survey that includes validated tools like the **Patient Satisfaction Questionnaire (PSQ-18)** and custom questions specific to the AI-enabled DR screening program. The survey will use a **Likert scale** (e.g., 1 = very dissatisfied to 5 = very satisfied) to evaluate domains such as accessibility, process experience, and outcome satisfaction. Surveys will be administered after screening and follow-up visits, either digitally or on paper, to capture immediate and longitudinal feedback. Responses will be anonymized and analyzed to assess overall satisfaction and identify areas for improvement.

## Cost-effectiveness analysis

The evidence on cost-effectiveness of screening for DR using digital technologies is mixed. For instance Avidor et al. found that the results varied from not being cost-effective to being cost-effective to being cost-saving based on the context and the frequency and modality of the screening and the use of telemedicine [35]. Given the resource constraints in an LMIC like India, it is imperative that the AI solutions proposed are cost-effective (preferably cost-saving) if they are to be adopted at scale.

We will calculate the incremental costs incurred at the diabetes clinic to set up the systematic screening program and an integrated referral pathway. This comprises the cost of additional personnel, if any, and amortisation of the equipment and the cost of personnel involved in grading of the images. Additional cost components will include personnel training and capacity building at the clinic level, costs of AI model integration and periodic validation, maintenance and software updates for AI and imaging devices, and costs associated with internet and data management infrastructure. We will also model the improvement in clinical outcomes due to the earlier diagnosis of DR based on standard disease progression Markov models. Finally, we will integrate the findings from the disease progression and costing analyses to calculate the overall cost-effectiveness of our intervention in terms of additional costs incurred per incremental QALY gained or DALY averted.

## Process evaluation

The process evaluation will focus on several key metrics to assess the effectiveness and efficiency of the AI-enabled systematic screening program. These metrics include the DR detection rate and STDR detection rate, which will measure the program's ability to identify diabetic retinopathy and sight-threatening diabetic retinopathy, respectively. Visual outcomes, specifically visual impairment (VI) and blindness, will be tracked to evaluate the impact of the program on patient health. The timely referral rate for screening, defined as referrals made within two weeks of contact with eligible PWD for DR screening, and the timely referral rate for management, defined as referrals made within two weeks of contact with eligible PWD diagnosed with STDR, will be measured to ensure prompt follow-up and treatment. Additionally, the treatment uptake rate and follow-up rates will be monitored to gauge patient adherence to recommended care. Lastly, a cost-effectiveness analysis will be conducted, considering both direct and indirect expenses, to determine the financial viability and benefits of the program.

## Program outcomes

The program's outcomes will be evaluated through metrics such as screening uptake rate, sensitivity, and specificity of AI versus manual system, ROC Curve for AI Diagnostic Accuracy for diagnosis of DR, referral rate, cost-effectiveness, patient and provider satisfaction, follow-up rates, and long-term outcomes. These outcomes will help determine the program's impact on patient health and healthcare costs. Due to the formatting constraints of the CTRI registry, certain fields were filled in a specific way, which might lead to minor discrepancies between the reported outcomes in the manuscript and the CTRI registry. (Table 1 shows the outcome measurement indicators for the project.).

## Ethics

The proposed study will comply with the ethical principles of research. The study will commence following approval and clearance from the Institutional Ethics Committees (IEC) of LV Prasad Eye Institute and IDEA Clinics. The Ethics Committee of LV Prasad Eye Institute approved the study on 06/10/2023, and the Tanvir Hospital Institutional Ethics Committee for Biomedical and Health Research granted approval on 07/11/2023.Any protocol amendments/modifications will be implemented only after Institutional Ethics Committee clearance. Informed written consent will be sought from all the patients after providing the necessary information regarding the study objectives as well as potential benefits and risks of participating in the study. They will also be informed that they can withdraw from the study at any point prior to, during or after the data collection. They will have the freedom to ask any queries pertaining to any aspect of the study. Confidentiality will be maintained and all the information regarding the participants will be protected under all circumstances. At LVPEI, The patients written consent for comprehensive examination done at LVPEI will be considered as study consent form.

**Table 1. Outcome measurement indicators: SMART DROP project.**

| Outcome | Definition | Numerator | Denominator |
|---|---|---|---|
| Screening uptake rate | Proportion of eligible patients who underwent DR screening out of the total eligible population. | Number of eligible patients who underwent DR screening | Total number of eligible patients in the population |
| Sensitivity of AI system | Proportion of true positive DR cases correctly identified by the AI system. | Number of true positive DR cases identified by the AI system | Total number of true DR cases confirmed by manual grading |
| Specificity of AI system | Proportion of true negative cases correctly identified by the AI system. | Number of true negative DR cases identified by the AI system | Total number of non-DR cases confirmed by manual grading |
| ROC curve for AI diagnostic accuracy | A graphical plot showing the diagnostic ability of the AI system by plotting true positive rate (sensitivity) against false positive rate (1-specificity). | True positive rate (sensitivity) | False positive rate (1-specificity) |
| Referral rate | Proportion of screened patients referred for further examination or treatment. | Number of patients referred for further examination or treatment | Total number of patients referred |
| Cost effectiveness | Ratio of the total costs associated with the screening program to the health outcomes achieved, typically measured as cost per quality-adjusted life year (QALY). | Total costs of the screening program | Total health outcomes achieved (e.g., QALYs gained, cases of blindness prevented) |
| Patient satisfaction | Proportion of patients reporting satisfaction with the DR screening process and outcomes. | Number of patients reporting satisfaction | Total number of patients who completed the satisfaction survey |
| Provider satisfaction | Proportion of healthcare providers reporting satisfaction with the AI-assisted DR screening program. | Number of providers reporting satisfaction | Total number of providers who completed the satisfaction survey |
| Follow-up rate | Proportion of patients who return for follow-up visits after initial DR screening or referral. | Number of patients who return for follow-up visits | Total number of patients advised to return for follow-up visits |
| Long-term outcomes | Proportion of patients with improved visual outcomes or stable DR over a defined period after screening or treatment. | Number of patients with improved visual outcomes or stable DR over time | Total number of patients tracked for long-term outcomes |

All participant data will be anonymized to ensure confidentiality. Unique identification codes will be assigned to participants, and no personally identifiable information (PII) will be linked directly to the dataset. Data will be securely stored in encrypted digital formats on password-protected servers with access restricted to authorized personnel. Hard copies, if any, will be stored in locked cabinets within secure facilities. Any data transfer between study sites will be conducted using secure, encrypted channels to maintain data security.

Participants who withdraw from the study will have their data excluded from subsequent analyses unless explicit consent is provided for its continued use. Patients lost to follow-up will be included in an intention-to-treat analysis to minimize bias and preserve the benefits of systematic recruitment. Sensitivity analyses will be performed to evaluate the potential impact of missing data. Where applicable, missing data will be addressed using appropriate imputation methods, such as multiple imputation or the last observation carried forward, to ensure the robustness and validity of the study findings.

## Discussion

The study will have a significant impact by establishing a referral pathway that integrates disparate parts of the private healthcare system, facilitating the early diagnosis of chronic conditions such as diabetic retinopathy (DR). The long-term goal is to create a cost-effective pathway for the management of DR. We plan to publish these findings in top peer-reviewed journals and present them at reputable conferences. Finally, the participating healthcare institutions (LVPEI and IDEA clinics) are committed to institutionalizing the pathway into their routine operations to provide higher-quality care to their patients.

### Strengths of the study

• Establishment of a systematic DR screening program at physician clinics, improving early detection and management of diabetic retinopathy.

• Utilization of various fundus cameras to evaluate the scalability, adaptability, and accuracy of the DR screening system.

• Development of an effective imaging pathway to enhance accuracy and reduce ungradable images.

• Completion of the referral pathway and creation of a universally implementable tracking system.

• Analysis of the economic impact, cost-effectiveness, and environmental impact of tele-DR screening.

### Limitations of the study

• The study is conducted in a private healthcare setup where costs are involved, which may result in the exclusion of some patients who could not afford the screening, potentially leading to missed enrolments.

• The use of non-mydriatic fundus photography may limit the quality of imaging due to undilated pupils, potentially affecting the accuracy of DR detection.

• The study is not designed as a randomized controlled trial (RCT), which would have required the involvement of multiple health facilities and diabetes clinics to generalize the findings more broadly.

The implementation of the study may face challenges such as variability in AI performance, patient adherence to referrals and follow-up, operational issues across multiple sites, patient acceptance of AI tools, and resource constraints for scale-up. To address these potential challenges, we plan periodic recalibration of the AI system, standardized technician training, and regular quality audits to ensure consistent performance. Technicians will undergo periodic training programs focused on image acquisition techniques, device handling, and identification of common image quality issues. In parallel, we will maintain regular communication with the AI solution providers to incorporate

necessary adjustments in the algorithm, particularly to optimize performance based on real-world sensitivity and specificity data. This dual approach is expected to enhance both image quality and the diagnostic accuracy of the AI system over time. Strategies to improve patient adherence will include employing navigators, simplifying referral pathways, and providing education on the importance of timely care. Operational issues will be managed through centralized coordination, cloud-based data management, and regular equipment maintenance. Patient trust in AI tools will be fostered through education, validation of AI outputs by experts, and sharing of success stories. Resource constraints will be mitigated by phased implementation in high-burden areas and leveraging public-private partnerships.

Additionally, ensuring adherence to the **Standard Operating Procedure (SOP)** in a private clinic chain—where business performance may take precedence—can be challenging, as operational priorities may not always align with clinical protocols. Furthermore, monitoring compliance across multiple locations is complex due to variations in workflow, staff engagement, and resource availability. Adherence to SOPs in a private clinic chain with business priorities will be ensured by aligning protocols with clinic goals, incentivizing compliance, and integrating them into existing workflows. Stakeholder engagement and automated compliance tracking will further support adherence. Monitoring across multiple locations will be managed through real-time dashboards, automated alerts, remote audits, and periodic site visits. Local champions will be designated in each clinic to enhance accountability and streamline implementation. These measures are expected to support effective and scalable implementation of the study.

Given that private providers manage 70–80% of all patients, with an even higher proportion in advanced ophthalmologic conditions like DR, the findings are highly relevant to real-world care delivery. The study's reliance on private healthcare clinics may limit generalizability by excluding patients unable to afford screening. To reduce selection bias, future studies should include public healthcare settings. Non-mydriatic fundus photography may impact image quality and DR detection accuracy; this can be mitigated through enhanced technician training, image quality audits, and advanced AI algorithms. These steps can improve the applicability of findings across diverse settings.

## Conclusion

The study findings have the potential to impact national programs like Ayushman Bharat by demonstrating the feasibility of integrating AI-enabled diabetic retinopathy screening into routine care. Evidence on accuracy, cost-effectiveness, and optimized referral pathways could enhance resource utilization and accessibility. Additionally, the results may inform policy decisions on reimbursing AI-assisted screening by trained non-medical technicians, contributing to broader coverage and earlier detection of DR nationwide.

## Acknowledgments

ICMR (Indian Council of Medical Research) (Indian Council of Medical Research)

## Author contributions

**Conceptualization:** Padmaja Kumari Rani, Sarang Deo.

**Data curation:** Padmaja Kumari Rani.

**Formal analysis:** Padmaja Kumari Rani.

**Funding acquisition:** Padmaja Kumari Rani, Raja Narayanan.

**Investigation:** Padmaja Kumari Rani, DurgaBhavani Kalavalapalli, Raja Narayanan, Shyam Kalavalapalli, Ritesh Narula, Rakesh K Sahay, Sarang Deo.

**Methodology:** Padmaja Kumari Rani, DurgaBhavani Kalavalapalli, Raja Narayanan, Shyam Kalavalapalli, Ritesh Narula, Rakesh K Sahay, Sarang Deo.

**Project administration:** Padmaja Kumari Rani, DurgaBhavani Kalavalapalli, Sarang Deo.

**Resources:** Padmaja Kumari Rani.

**Supervision:** Padmaja Kumari Rani.

**Validation:** Padmaja Kumari Rani.

**Visualization:** Padmaja Kumari Rani.

**Writing – original draft:** Padmaja Kumari Rani.

**Writing – review & editing:** Padmaja Kumari Rani, DurgaBhavani Kalavalapalli, Raja Narayanan, Shyam Kalavalapalli, Ritesh Narula, Rakesh K Sahay, Sarang Deo.

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
