## [Decision Letter · Decision Letter 0]

19 Jan 2025

PONE-D-24-54914SMART (artificial intelligence enabled) DROP (Diabetic Retinopathy Outcomes and Pathways): Study Protocol for Diabetic Retinopathy ManagementPLOS ONE

Dear Dr. Rani,

Thank you for submitting your manuscript to PLOS ONE. After careful consideration, we feel that it has merit but does not fully meet PLOS ONE’s publication criteria as it currently stands. Therefore, we invite you to submit a revised version of the manuscript that addresses the points raised during the review process.

We look forward to receiving your revised manuscript.

Kind regards,

Kumar Saurabh

Academic Editor

PLOS ONE

Journal Requirements:

4. We note that Figure 2 in your submission contain map/satellite images which may be copyrighted. All PLOS content is published under the Creative Commons Attribution License (CC BY 4.0), which means that the manuscript, images, and Supporting Information files will be freely available online, and any third party is permitted to access, download, copy, distribute, and use these materials in any way, even commercially, with proper attribution. For these reasons, we cannot publish previously copyrighted maps or satellite images created using proprietary data, such as Google software (Google Maps, Street View, and Earth). For more information, see our copyright guidelines: http://journals.plos.org/plosone/s/licenses-and-copyright.

5. Please ensure that you refer to Figure 1 in your text as, if accepted, production will need this reference to link the reader to the figure.

6. Please include a separate caption for each figure in your manuscript.

Additional Editor Comments:

Dear Authors

The study protocol described in the manuscript requires further specifications to make it more objective. Please go through the reviewers' comments and modify the manuscript accordingly.

Reviewers' comments:

Reviewer's Responses to Questions

**Comments to the Author**

1. Does the manuscript provide a valid rationale for the proposed study, with clearly identified and justified research questions?

Reviewer #1: Yes

Reviewer #2: Partly

Reviewer #3: Yes

2. Is the protocol technically sound and planned in a manner that will lead to a meaningful outcome and allow testing the stated hypotheses?

Reviewer #1: Yes

Reviewer #2: Yes

Reviewer #3: Yes

3. Is the methodology feasible and described in sufficient detail to allow the work to be replicable?

Reviewer #1: Yes

Reviewer #2: Yes

Reviewer #3: Yes

4. Have the authors described where all data underlying the findings will be made available when the study is complete?

Reviewer #1: No

Reviewer #2: Yes

Reviewer #3: No

5. Is the manuscript presented in an intelligible fashion and written in standard English?

Reviewer #1: Yes

Reviewer #2: Yes

Reviewer #3: Yes

6. Review Comments to the Author

You may also provide optional suggestions and comments to authors that they might find helpful in planning their study.

Reviewer #1: The study is relevant and addresses a critical public health issue by focusing on early detection of diabetic retinopathy (DR) and the potential impact of systematic screening.

Objectives are clearly stated and include outcomes such as earlier detection, reduced visual impairment, and a comparison of different screening settings.

The inclusion of AI accuracy as a secondary outcome is novel and could inform future AI implementations in healthcare.

Sample Size and Sampling

• The sample size appears to be based on prevalence but may not account for the need to evaluate AI diagnostic performance (sensitivity/specificity). Please include a separate calculation for diagnostic evaluation.

• Clarify the sampling method (e.g., random or convenience sampling) and ensure it minimizes bias. For example, confirm whether recruitment will be balanced across physician clinics and eye care settings.

Statistical Analysis:

Provide more details on the statistical methods to evaluate key outcomes.

• Use sensitivity, specificity, and receiver operating characteristic (ROC) curve analysis to validate the AI tool.

• Include multivariable models to adjust for confounders like age, diabetes duration, and baseline visual acuity.

Ethical Considerations:

• Describe how participant data will be anonymized and securely managed.

• Detail how patients who withdraw or are lost to follow-up will be handled in the analysis.

Reviewer #2: I have several concerns regarding the study's methodology and statistical approach:

I am not clear about rationale for the association between the intervention and improved AI prediction accuracy. Which AI model will be employed, and how will the authors test and validate its performance?

In line 213, it seems that no technician will confirm or rule out an AI-determined ‘no DR diagnosis.’ If this is the case, how will the specificity of the model be evaluated?

Line 242, is N=25 sufficient? How was this sample size determined?

The statistical analysis appears to address only one outcome. How will other outcomes be accounted for, and which statistical tests will be used for group comparisons?

What method was used to calculate the sample size, and what effect size was assumed in this calculation?

Reviewer #3: The study addresses a critical public health challenge, particularly in low-resource settings like India, where the burden of diabetes and DR continues to rise. By leveraging innovative technologies and multidisciplinary collaboration, the proposed study has potential to significantly improve access to timely DR diagnosis, reduce visual impairment and inform cost effective health care strategies. Overall, the manuscript is well structured and demonstrates a strong commitment to research rigor and public health relevance. Following are areas that could benefit from further elaboration:

Introduction:

1. Consider emphasizing the significance of cost effectiveness more explicitly as that is one of the study objectives.

Methods:

2. Include a brief justification for the choice of study sites, addressing their relevance and representativeness for scaling the findings.

3. Provide information on the AI system, such as its development, training dataset characteristics and prior accuracy metrics, if available.

4. Expand on how patient satisfaction will be assessed including details on survey tools or scales to be used.

5. Provide a brief overview of data protection measures, including encryption methods, user access protocols and compliance with relevant data protection regulations.

Discussion:

6. Critically analyse potential challenges in implementation, such as variability in AI performance or patient adherence, and discuss possible mitigation strategies.

7. Highlight the policy level implications of the study findings, particularly how they could influence national programs such as Ayushman Bharat.

8. The authors have just listed the limitations of the study, which is important. Consider also discussing how these limitations might affect the generalizability of the findings. Additionally, suggest ways to minimise the potential selection biases introduced by including only private clinics and address image quality issues due to reliance on non-mydriatic cameras.

7. PLOS authors have the option to publish the peer review history of their article (what does this mean? ). If published, this will include your full peer review and any attached files.

**Do you want your identity to be public for this peer review?** For information about this choice, including consent withdrawal, please see our Privacy Policy .

Reviewer #1: No

Reviewer #2: No

Reviewer #3: **Yes: ** Sanil Joseph

---

## [Author Response · Author response to Decision Letter 0]

26 Feb 2025

Response: Suggested changes have been made in the revised manuscript as per the guidelines.

Response: Suggested changes have been made in the revised manuscript.

Response: Suggested changes have been made in the revised manuscript.

4. We note that Figure 2 in your submission contain map/satellite images which may be copyrighted. All PLOS content is published under the Creative Commons Attribution License (CC BY 4.0), which means that the manuscript, images, and Supporting Information files will be freely available online, and any third party is permitted to access, download, copy, distribute, and use these materials in any way, even commercially, with proper attribution. For these reasons, we cannot publish previously copyrighted maps or satellite images created using proprietary data, such as Google software (Google Maps, Street View, and Earth). For more information, see our copyright guidelines: http://journals.plos.org/plosone/s/licenses-and-copyright.

Response: We have updated Figure 2 to use OpenStreetMap, which is ODbL-licensed and meets PLOS requirements. The figure now includes proper attribution: "Map data © OpenStreetMap contributors, licensed under ODbL."

5. Please ensure that you refer to Figure 1 in your text as, if accepted, production will need this reference to link the reader to the figure.

Response: Suggested changes have been made in the revised manuscript.

Figure 1 is supporting information file (SPIRIT Template for Clinical Trial) for the study. This has been cited in the revised manuscript under supporting files list.

6. Please include a separate caption for each figure in your manuscript.

Response: Suggested changes have been made in the revised manuscript.

Response: Suggested changes have been made in the revised manuscript.

Additional Editor Comments:

Dear Authors

The study protocol described in the manuscript requires further specifications to make it more objective. Please go through the reviewers' comments and modify the manuscript accordingly.

Response: We thank the reviewers for their valuable comments, all of which have been addressed in the revised manuscript.

PONE-D-24-54914

SMART (artificial intelligence enabled) DROP (Diabetic Retinopathy Outcomes and Pathways): Study Protocol for Diabetic Retinopathy Management

PLOS ONE

Reviewer #1: The study is relevant and addresses a critical public health issue by focusing on early detection of diabetic retinopathy (DR) and the potential impact of systematic screening.

Objectives are clearly stated and include outcomes such as earlier detection, reduced visual impairment, and a comparison of different screening settings.

The inclusion of AI accuracy as a secondary outcome is novel and could inform future AI implementations in healthcare.

Sample Size and Sampling

• The sample size appears to be based on prevalence but may not account for the need to evaluate AI diagnostic performance (sensitivity/specificity). Please include a separate calculation for diagnostic evaluation.

Response: We have included separate sample size calculation for diagnostic performance of AI in the revised manuscript. (Page no:15 Line no:337-342 )

• Clarify the sampling method (e.g., random or convenience sampling) and ensure it minimizes bias. For example, confirm whether recruitment will be balanced across physician clinics and eye care settings.

Response: The study will employ a convenience sampling method, recruiting all consecutive eligible individuals with diabetes mellitus (DM) visiting physician clinics and eye care settings who meet the standard inclusion and exclusion criteria and provide informed consent. While convenience sampling is utilized, potential sources of bias include selection bias, site-specific variation, and differential access to care. To mitigate these, systematic recruitment will be ensured by enrolling all eligible patients consecutively at each site, reducing the likelihood of preferential selection. Recruitment will take place across multiple physician clinics and eye care settings to improve representativeness and minimize site-specific variation. Differential access to care will be monitored by assessing demographic and clinical characteristics of enrolled participants, allowing adjustments to recruitment strategies if needed. To prevent observer bias, standardized screening protocols and training for healthcare personnel will be implemented, ensuring consistency in recruitment and data collection. These measures aim to enhance the validity and generalizability of the findings while acknowledging the inherent limitations of convenience sampling.

Above clarification has been included in the revised text.(Page no:13-14, Line no: 307-321)

Statistical Analysis:

Provide more details on the statistical methods to evaluate key outcomes.

• Use sensitivity, specificity, and receiver operating characteristic (ROC) curve analysis to validate the AI tool.

Response: We have included above parameters for estimating diagnostic accuracy of AI. The table has been updated to include ROC outcome measure for diagnostic performance in the revised manuscript. Accuracy of AI-enabled cameras: Taking the grading of images done by the experts as the ground truth, we will estimate the accuracy of the AI- enabled cameras (operated by a non- ophthalmic technician) using standard measures such as sensitivity, specificity, and area under the ROC (Receiver Operating Characteristic). (Page no:17 Line no:395 and Page 18,Table )

• Include multivariable models to adjust for confounders like age, diabetes duration, and baseline visual acuity.

Reply: Suggested changes have been incorporated in the revised manuscript. (Page no: 13,Line no: 298-306)Multivariable regression models will be employed to adjust for potential confounders, including age, diabetes duration, baseline visual acuity, and other relevant factors. These models will allow for a more accurate assessment of the relationships between the intervention and key outcomes, such as disease staging at diagnosis, referral rates, and long-term visual outcomes, by accounting for the influence of these confounders. Both linear and logistic regression models will be used, depending on the nature of the outcome variable, to ensure robust and reliable results.

Ethical Considerations:

• Describe how participant data will be anonymized and securely managed.

• Detail how patients who withdraw or are lost to follow-up will be handled in the analysis.

Reply: Suggested changes have been incorporated in the revised text. (Page no:19-20, Line no: 412-433).All participant data will be anonymized to ensure confidentiality. Unique identification codes will be assigned to participants, and no personally identifiable information (PII) will be linked directly to the dataset. Data will be securely stored in encrypted digital formats on password-protected servers with access restricted to authorized personnel. Hard copies, if any, will be stored in locked cabinets within secure facilities. Any data transfer between study sites will be conducted using secure, encrypted channels to maintain data security.

Participants who withdraw from the study will have their data excluded from subsequent analyses unless explicit consent is provided for its continued use. Patients lost to follow-up will be included in an intention-to-treat analysis to minimize bias and preserve the benefits of systematic recruitment. Sensitivity analyses will be performed to evaluate the potential impact of missing data. Where applicable, missing data will be addressed using appropriate imputation methods, such as multiple imputation or the last observation carried forward, to ensure the robustness and validity of the study findings.

Reviewer #2: I have several concerns regarding the study's methodology and statistical approach:

I am not clear about rationale for the association between the intervention and improved AI prediction accuracy. Which AI model will be employed, and how will the authors test and validate its performance?

Response: The project employs two fundus camera models with inbuilt AI algorithms. Non-mydriatic fundus photography of both eyes will be performed using AI-integrated retinal cameras (Forus, 3 Nethra Classic HD with AI Integration in two facilities, and Remidio, Fundus on Phone with AI integration, in two facilities). All images are sent by cloud transfer to a central image reading centre, where in the images are manually graded. Taking the grading of images done by the experts at the image reading centre as the ground truth, we will estimate the accuracy of the AI- enabled cameras (operated by a non- ophthalmic technician) using standard measures such as sensitivity, specificity, and area under the ROC (Receiver Operating Characteristic). (Page no:9 Line no:210-215, Page no:17 Line no:395 and Page 18,Table )

In line 213, it seems that no technician will confirm or rule out an AI-determined ‘no DR diagnosis.’ If this is the case, how will the specificity of the model be evaluated?

Response: The proposed methodology ensures timely referral of patients and aims to minimize false positive referrals by the AI system. To evaluate the specificity of the model, all fundus images will be graded at a central image reading center by expert graders. The manual grading results from the central reading center will serve as the ground truth to accurately determine the sensitivity and specificity of the AI system for detecting diabetic retinopathy. This approach ensures that the performance metrics of the AI model are validated against a reliable and expert-verified standard.

Line 242, is N=25 sufficient? How was this sample size determined?

Response: The visual acuity tool used in the study (Peek Vision) has been globally validated in multiple studies and effectively utilized by technicians and field workers in diverse settings.[33, 34]The sample size of N=25 patients (50 eyes) is intended for pilot testing following standardized training. This pilot will serve as an initial assessment of the reliability and accuracy of visual acuity estimation using the mobile VA application by trained technicians. The pilot testing results will be used to validate the training protocol and identify any gaps requiring refinement.

To ensure ongoing reliability, periodic audits will be conducted during training refresher programs, where technicians' performance will be re-evaluated. Additionally, comparisons of visual acuity estimates from referred patients will be performed to assess consistency over time. This iterative approach ensures that the quality of visual acuity measurements is maintained throughout the study. (Page no:11,Line no: 253-264)

The statistical analysis appears to address only one outcome. How will other outcomes be accounted for, and which statistical tests will be used for group comparisons?

Response: The suggested changes have been included in the revised manuscript.

Sample size for other outcomes such as AI diagnostic accuracy and patient satisfaction has been included in the revised text. Page no:15, Line no:337-358 )

Multivariable regression models will be employed to adjust for potential confounders, including age, diabetes duration, baseline visual acuity, and other relevant factors. These models will allow for a more accurate assessment of the relationships between the intervention and key outcomes, such as disease staging at diagnosis, referral rates, and long-term visual outcomes, by accounting for the influence of these confounders. Both linear and logistic regression models will be used, depending on the nature of the outcome variable, to ensure robust and reliable results. Additional statistical tests, such as chi-square tests for categorical outcomes and t-tests or ANOVA for continuous outcomes, will be employed as appropriate. (Page no:13, Line no:298-306 )

What method was used to calculate the sample size, and what effect size was assumed in this calculation?

Response: The sample size was calculated using a difference in proportions method to detect meaningful differences in the staging of sight-threatening diabetic retinopathy (STDR) between the intervention and standard-of-care (SOC) arms at their first ophthalmology visit. An effect size was assumed based on historical data, with an estimated 50% prevalence of STDR in the SOC arm and a lower prevalence of approximately 10% in the intervention arm at the time of initial screening. However, to account for non-compliance, the prevalence of STDR among patients completing their first ophthalmology visit in the intervention arm was adjusted to 25%. The calculat362-367ion assumed a power of 80% and a Type-I error rate of 5%.

Reviewer #3: The study addresses a critical public health challenge, particularly in low-resource settings like India, where the burden of diabetes and DR continues to rise. By leveraging innovative technologies and multidisciplinary collaboration, the proposed study has potential to significantly improve access to timely DR diagnosis, reduce visual impairment and inform cost effective health care strategies. Overall, the manuscript is well structured and demonstrates a strong commitment to research rigor and public health relevance. Following are areas that could benefit from further elaboration:

Introduction:

1. Consider emphasizing the significance of cost effectiveness more explicitly as that is one of the study objectives.

Response: The suggested changes have been included in the revised manuscript.(Page no: 16,Line no: 362-367 )The evidence on cost-effectiveness of screening for DR using digital technologies is mixed. For instance Avidor et al. (2020) found that the results varied from not being cost-effective to being cost-effective to being cost-saving based on the context and the frequency and modality of the screening and the use of telemedicine. Given the resource constraints in an LMIC like India, it is imperative that the AI solutions proposed are cost-effective (preferably cost-saving) if they are to be adopted at scale.

Methods:

2. Include a brief justification for the choice of study sites, addressing their relevance and representativeness for scaling the findings.

Response: The suggested changes have been included in the revised manuscript.(Page no:8,Line no: 177-185 ).

The LV Prasad Eye Institute is a leading tertiary eye care center with advanced infrastructure for diabetic retinopathy diagnosis and treatment, making it an ideal reference for validating the intervention. The IDEA Clinics are strategically located in urban areas with a high burden of diabetes and diabetic retinopathy, ensuring the recruitment of a relevant and representative population. These clinics provide comprehensive diabetes care and offer an optimal setting to test the integration of AI-enabled diabetic retinopathy screening within existing care pathways. The combinat

---

## [Decision Letter · Decision Letter 1]

8 Apr 2025

PONE-D-24-54914R1

SMART (artificial intelligence enabled) DROP (Diabetic Retinopathy Outcomes and Pathways): Study Protocol for Diabetic Retinopathy Management

PLOS ONE

Dear Dr. Rani,

Thank you for submitting your manuscript to PLOS ONE. After careful consideration, we feel that it has merit but does not fully meet PLOS ONE’s publication criteria as it currently stands. Therefore, we invite you to submit a revised version of the manuscript that addresses the points raised during the review process.

We look forward to receiving your revised manuscript.

Kind regards,

Kumar Saurabh

Academic Editor

PLOS ONE

Journal Requirements:

Reviewers' comments:

Reviewer's Responses to Questions

**Comments to the Author**

1. Does the manuscript provide a valid rationale for the proposed study, with clearly identified and justified research questions?

Reviewer #1: Yes

Reviewer #2: Yes

2. Is the protocol technically sound and planned in a manner that will lead to a meaningful outcome and allow testing the stated hypotheses?

Reviewer #1: Yes

Reviewer #2: Yes

3. Is the methodology feasible and described in sufficient detail to allow the work to be replicable?

Reviewer #1: Yes

Reviewer #2: Yes

4. Have the authors described where all data underlying the findings will be made available when the study is complete?

Reviewer #1: Yes

Reviewer #2: Yes

5. Is the manuscript presented in an intelligible fashion and written in standard English?

Reviewer #1: Yes

Reviewer #2: Yes

6. Review Comments to the Author

You may also provide optional suggestions and comments to authors that they might find helpful in planning their study.

Reviewer #1: The revisions have significantly improved the clarity and methodological rigor of your manuscript. The inclusion of a separate sample size calculation for evaluating AI diagnostic performance strengthens the study’s statistical foundation.

The addition of sensitivity, specificity, and ROC curve analysis for AI validation ensures a more robust assessment of its effectiveness.

The revised statistical analysis section now includes appropriate multivariable regression models to adjust for confounders, improving the validity of findings.

While cost-effectiveness has been emphasized, consider including a clearer breakdown of expected cost components (e.g., personnel training, AI maintenance) to strengthen this section.

Image Quality in Non-Mydriatic Photography: While the manuscript acknowledges limitations related to image quality, additional discussion on how AI adjustments or technician training will mitigate this issue would be beneficial.

Reviewer #2: All my comments are addressed.

7. PLOS authors have the option to publish the peer review history of their article (what does this mean? ). If published, this will include your full peer review and any attached files.

**Do you want your identity to be public for this peer review?** For information about this choice, including consent withdrawal, please see our Privacy Policy .

Reviewer #1: **Yes: ** R Janani Surya

Reviewer #2: No

---

## [Author Response · Author response to Decision Letter 1]

16 Apr 2025

Response to Reviewers: We thank reviewers for their valuable comments. All suggestions have been incorporated in the revised manuscript.

Reviewer #1: The revisions have significantly improved the clarity and methodological rigor of your manuscript. The inclusion of a separate sample size calculation for evaluating AI diagnostic performance strengthens the study’s statistical foundation.

The addition of sensitivity, specificity, and ROC curve analysis for AI validation ensures a more robust assessment of its effectiveness.

Response : Thank you for your valuable comments.

The revised statistical analysis section now includes appropriate multivariable regression models to adjust for confounders, improving the validity of findings.

While cost-effectiveness has been emphasized, consider including a clearer breakdown of expected cost components (e.g., personnel training, AI maintenance) to strengthen this section.

Response : Thank you. Suggested changes have been included in the revised manuscript. Page 16, Lines 372-375.

Image Quality in Non-Mydriatic Photography: While the manuscript acknowledges limitations related to image quality, additional discussion on how AI adjustments or technician training will mitigate this issue would be beneficial.

Response : Thank you. Suggested changes have been included in the revised manuscript. Page 21,Line 471-477

Reviewer #2: All my comments are addressed.

Response : Thank you for your valuable comments.

---

## [Decision Letter · Decision Letter 2]

24 Apr 2025

SMART (artificial intelligence enabled) DROP (Diabetic Retinopathy Outcomes and Pathways): Study Protocol for Diabetic Retinopathy Management

PONE-D-24-54914R2

Dear Dr. Rani,

We’re pleased to inform you that your manuscript has been judged scientifically suitable for publication and will be formally accepted for publication once it meets all outstanding technical requirements.

Kind regards,

Kumar Saurabh

Academic Editor

PLOS ONE

Additional Editor Comments (optional):

The manuscript can be accepted in present form.

Reviewers' comments:

Reviewer's Responses to Questions

**Comments to the Author**

1. Does the manuscript provide a valid rationale for the proposed study, with clearly identified and justified research questions?

Reviewer #2: Yes

2. Is the protocol technically sound and planned in a manner that will lead to a meaningful outcome and allow testing the stated hypotheses?

Reviewer #2: Yes

3. Is the methodology feasible and described in sufficient detail to allow the work to be replicable?

Reviewer #2: Yes

4. Have the authors described where all data underlying the findings will be made available when the study is complete?

Reviewer #2: Yes

5. Is the manuscript presented in an intelligible fashion and written in standard English?

Reviewer #2: Yes

6. Review Comments to the Author

You may also provide optional suggestions and comments to authors that they might find helpful in planning their study.

Reviewer #2: All my concerns are addressed.

7. PLOS authors have the option to publish the peer review history of their article (what does this mean? ). If published, this will include your full peer review and any attached files.

**Do you want your identity to be public for this peer review?** For information about this choice, including consent withdrawal, please see our Privacy Policy .

Reviewer #2: No

---

## [Editor Report · Acceptance letter]

PONE-D-24-54914R2

PLOS ONE

Dear Dr. Rani,

I'm pleased to inform you that your manuscript has been deemed suitable for publication in PLOS ONE. Congratulations! Your manuscript is now being handed over to our production team.

Kind regards,

on behalf of

Dr. Kumar Saurabh

Academic Editor

PLOS ONE